# Predicting harmful alcohol use prevalence in Sub-Saharan Africa between 2015 and 2019: Evidence from population-based HIV impact assessment

**Mtumbi Goma** [1]*, **Wingston Felix Ng'ambi** [1,2], **Cosmas Zyambo** [1]

**1** Department of Community Health and Family Medicine, University of Zambia, Lusaka, Zambia, **2** Health Economics and Policy Unit, Department of Health Systems and Policy, Kamuzu University of Health Sciences, Lilongwe, Malawi

* mtumbirobertgoma@gmail.com

**Data Availability Statement:** The data utilized in this manuscript were sourced from the International Center for AIDS Care and Treatment Programs (ICAP) at Columbia University -

## Abstract

### Introduction

Harmful alcohol use is associated with significant risks to public health outcomes worldwide. Although data on harmful alcohol use have been collected by population-based HIV Impact Assessment (PHIA), there is a dearth of analysis on the effect of HIV/ART status on harmful alcohol use in the Sub-Saharan Africa (SSA) countries with PHIA surveys.

### Methods

A secondary analysis of the PHIA surveys: Namibia (n = 27,382), Tanzania (n = 1807), Zambia (n = 2268), Zimbabwe (n = 3418), Malawi (n = 2098), Namibia (n = 27,382), and Eswatini (n = 2762). Using R version 4.2, we analysed the uptake and correlates of harmful alcohol consumption in SSA. The cutoff point for statistically significant was P<0.05.

### Results

Of the 12,460 persons, 15% used alcohol harmfully. Harmful alcohol use varied by countries and ranged from 8.7% in Malawi to 26.1% in Namibia (P<0.001). Being female or HIV-positive and on ART were associated with less-likelihood of harmful alcohol consumption however persons that were HIV-positive and not on ART was associated with higher likelihood of harmful alcohol use (OR = 1.49, 95% CI: 1.32–1.69, P<0.001). The best performing models were Lasso or Super Learner or Random Forest were the best performing models while gradient boosting models or sample mean did not perform well.

### Conclusion

Harmful alcohol use was high. Harmful alcohol use varied by countries, sex, age, HIV/ART status and marital status. Therefore, there is a need to introduce or enforce harmful alcohol use control policies in SSA through taking into account these characteristics.

Population-based HIV Impact Assessment (PHIA) website. The datasets are accessible at the following link: https://phia-data.icap.columbia.edu/datasets. Please refer to the data access policies of PHIA when accessing the datasets.

**Funding:** The author(s) received no specific funding for this work.

**Competing interests:** The authors have declared that no competing interests exist

## Introduction

Around 33.3 million deaths globally are attributed to harmful alcohol use each year, accounting for roughly 50.9 percent of all fatalities. Additionally, alcohol use is connected to 5% of the world's disease burden. Harmful alcohol use is defined as alcohol consumption that results in physical or psychological harm. The Harmful alcohol use is recognized by the World Health Organisation (WHO) 10th revision of the International Classification of Diseases (ICD-10) indicating that alcohol is responsible for physical or psychological harm, the nature of the harm is identifiable, alcohol consumption has persisted for at least 1 month or has occurred repeatedly over the previous 12-month period, and the individual does not meet the criteria for alcohol dependence [1, 2]. More than 200 health conditions have been found to be causally linked to alcohol consumption in recent times. This includes new research showing a connection between harmful alcohol use and the incidence and clinical outcomes of infectious diseases like tuberculosis, HIV/AIDS, and pneumonia [3]. In sub-Saharan Africa (SSA), where heavy alcohol consumption is common, harmful alcohol use is linked to disorders. According to estimates, the prevalence of alcohol use disorders is two to four times higher in people living with HIV (PLHIV) than in people without HIV infection. This increases the risk of HIV infection and transmission in a variety of contexts [4].

The harmful use of alcohol constitutes 5.1% of the worldwide burden of disease [5]. Notably, harmful alcohol use emerges as the primary risk factor for premature mortality and disability in adults, contributing to 10% of all deaths globally. Additionally, disadvantaged and particularly vulnerable populations face elevated rates of alcohol-related deaths and hospitalizations [6]. Harmful alcohol use is associated with health status. In a longitudinal cohort study conducted in the United States of America, involving individuals with HIV who were engaged in care across seven clinics participating in the Centers for AIDS Research Network of Integrated Care Systems between January 2011 and June 2014 (n = 5,046). The prevalence of heavy alcohol use was found to be 21% among women, 31% among men who have sex with women, and 37% among men who have sex with men. Among women, heavy alcohol use was associated with a subsequently decreased median self-reported health status score compared to those with no or moderate alcohol use [7].

In Ethiopia, the pooled prevalence of alcohol use among people PLWHIV was found to be high [8]. A study on alcohol use and abuse among rural adults was conducted in Zimbabwe found that harmful alcohol consumption correlated with being male, older, unmarried, more educated, of Shona ethnicity, frequent travelers, employed, those without religion, or residing in areas with specific demographic characteristics. Therefore, the study reaffirms the influence of socio-demographic and cultural factors on alcohol use, aligning with patterns seen in other countries [9]. The prevalence of harmful alcohol consumption presents a noteworthy concern especially within the population of PLHIVs, particularly among those who have been on ART for a relatively longer time [10, 11]. Disadvantaged and vulnerable populations in SSA face higher rates of alcohol-related deaths and hospitalizations, highlighting the urgency to understand the dynamics of harmful alcohol use especially in the context of HIV care in these communities [12]. Therefore, using data from population-based HIV Impact Assessment (PHIA), we conducted a study to predict harmful alcohol use, and comprehensively understand the relationship between harmful alcohol use and HIV or Antiretroviral Therapy (ART) status while considering the influence of various contributing factors. The following were the specific objectives to; (a) determine the prevalence of harmful alcohol use amongst adult persons living in sub-Saharan Africa, and (b) examine the relationship between harmful alcohol use and HIV/ART status amongst adult persons living in Sub-Saharan Africa.

## Methods

### Ethical considerations

The initial ethics approval for the study protocols were obtained from the Centers for Disease Control and Prevention Institutional Review Board (IRB), the Columbia University Medical Center IRB, and relevant local regulatory bodies. We obtained permission to use this data from the International Center for AIDS Care and Treatment Programs (ICAP) at Columbia University. The PHIA datasets were downloaded from https://phia-data.icap.columbia.edu/datasets. As this study used secondary anonymised data, individual informed consent is not required. Furthermore, to conduct this study, we obtained ethics approval from both the University of Zambia Research Ethics committee (UNZABREC Ref. No. 4842–2024) and Zambia National Research Authority (NRA).

### Study design and sampling

This study used secondary data from the Population-based HIV Impact Assessment (PHIA) collected from SSA between 2015 and 2019. All persons aged at least 15 years were included in the analysis. Since 2014, the PHIA Project conducts nationally representative surveys to capture the state of the HIV epidemic in the most-affected countries. This effort is led by the Ministry of Health in each participating country and funded by the U.S. President's Emergency Plan for AIDS Relief (PEPFAR) through the U.S. Centers for Disease Control and Prevention (CDC) with technical support from ICAP at Columbia University [13]. The PHIA utilized two-stage cluster design with census enumeration areas being the first stage to be sampled and the households being the second stage to be selected in order to achieve a representative sample [9, 10, 12]. The samples are stratified by rural and urban location in all the countries. The numbers of households to be included in the surveys vary from country to country. For example, 30 households were selected per EA, with a minimum of 15 households in Zambia, Zimbabwe and Malawi while a maximum of 60 in Zimbabwe and Malawi and 50 in Zambia [9].

### Data management

Data management was done in R version 4.2. We merged data from six Sub-Saharan countries namely Eswatini, Malawi, Namibia, Tanzania, Zambia, and Zimbabwe. The primary outcome variable of interest was 'harmful alcohol use'. 'Harmful alcohol use' in this context refers to patterns of drinking that result in adverse physical and psychological health outcomes [1]. 'Harmful alcohol use' was assessed using the Alcohol Use Disorder Identification Test–C (AUDIT-C) based on the frequency of drinking, drinking category and quantity of drinking. The AUDIT-C used the three standard questions from module 10 of the PHIA questionnaire to generate the scores [14]. The AUDIT-C is scored on a scale of 0–12 (scores of 0 reflect no alcohol use). Alcohol drinking is based on the AUDIT-C tool (AUDIT-C score $\leq 2$ for women and $\leq 3$ for men was classified as 'harmful alcohol use' (no); AUDIT-C score $\geq 3$ for women and $\geq 4$ for men was classified as 'harmful alcohol use' (yes).

The independent variables included age groups (in complete calendar years), sex (male/female), marital status (single, married, widowed and divorced), area of residence (rural/urban), highest education level (none, primary, secondary and tertiary), wealth index (poorest, poor, middle, rich and richest), country (stratified into Eswatini, Malawi, Namibia, Tanzania, Zambia and Zimbabwe), and HIV/ART status (negative, HIV+ not on ART and HIV- on ART).

## Data analysis

The analysis of the data was done in R v4.2. A descriptive analysis was first performed detailing the characteristics of the study population. We calculated the frequencies, proportions using stratification as appropriate. We also fitted univariable analysis of each of the independent variable and 'harmful alcohol use'. The multivariable logistic regression model was fitted using forward and backward model building techniques. In the multivariable model, age group, gender, marital status, HIV/ART status, area of residence, highest education level, wealth index and relationship to the household head were included in the model based on their P <0.05 at univariable analysis. The logistic regression outputs were reported both as unadjusted (cOR) and adjusted odds ratio (aOR). We fitted a univariable and multivariable logistic regression model of harmful alcohol use, with harmful alcohol use clustered by country.

We employed to test and apply machine learning (ML) methods through Super Learner, Decision Tree, Random Forest (RF), Lasso Regression, Sample mean and Gradient boosting in order to determine which of the models worked better or best in predicting 'harmful alcohol use'. We included in the ML model the predictor variables that were significant under the conventional statistical modelling. The researchers used supervised learning techniques to train the model on a variety of risk variables. When the algorithm was effectively trained, it could predict 'harmful alcohol use' when applied to new data. A classification algorithm is a form of model that generates discrete categories. We fitted on the training dataset (70% of the original dataset) and evaluated the models based on the test dataset (30% of the original dataset). Model evaluation was performed to see how well the classification model worked and how well it classified data. In this study, a confusion matrix and receiver operating characteristic (ROC) curve were used to assess algorithm performance using various metrics such as accuracy, sensitivity, specificity, F1 score, and area under the curve (AUC) [15, 16]. Statistical significance was set at p<0.05. The model used in the analysis considered only complete observation and all observations with missing data were excluded.

## Results

### Respondent characteristics

The characteristics of participants from the six countries are shown in Table 1. There were 14616 individuals. Of these; 23.3% were from Zimbabwe while 12.4% were from Tanzania, the majority were aged 35–39 years while the least were in the age group of 50–54 years, nearly two-third of the individuals were females, 49.9% of the individuals were married, 50.4% of the individuals were HIV positive and receiving ART, 62.2% resided in rural areas, the majority had primary school education while the minority had no education, most of the individuals were household heads while the least were children, and nearly 20% of the individuals were in each wealth index quintile. There was a similar pattern of the distribution of the characteristics with the general pattern by country.

### Prevalence of harmful alcohol use according to the country

Of the 14616 individuals, 2156 (14.8%) used alcohol in a harmful way. The prevalence of harmful alcohol use is shown in Table 1. The prevalence of harmful alcohol use was the least amongst the persons aged 15–19 and highest amongst those aged 25–54 years. Harmful alcohol use was more prevalent amongst: the males than females, those divorced than those widowed, those HIV positive and not on ART than those HIV positive and on ART, urban than rural residents, those with no education or those with tertiary education than their counterparts,

**Table 1. Sociodemographic characteristics of respondents in Eswatini, Malawi, Namibia, Tanzania, Zambia and Zimbabwe between 2015 and 2019.**

| Characteristics | Eswatini (N = 2762) | Malawi (N = 2098) | Namibia (N = 2263) | Tanzania (N = 1807) | Zambia (N = 2268) | Zimbabwe (N = 3418) | Overall[a] (N = 14616) | Harmful alcohol use prevalence[b] |
|---|---|---|---|---|---|---|---|---|
| **Age groups** | | | | | | | | |
| 15–19 | 369 (13.4%) | 222 (10.6%) | 286 (12.6%) | 183 (10.1%) | 319 (14.1%) | 381 (11.1%) | 1760 (12.0%) | 81 (4.6%) |
| 20–24 | 307 (11.1%) | 222 (10.6%) | 208 (9.2%) | 192 (10.6%) | 271 (11.9%) | 288 (8.4%) | 1488 (10.2%) | 181 (12.2%) |
| 25–29 | 343 (12.4%) | 262 (12.5%) | 242 (10.7%) | 192 (10.6%) | 273 (12.0%) | 298 (8.7%) | 1610 (11.0%) | 286 (17.8%) |
| 30–34 | 385 (13.9%) | 338 (16.1%) | 262 (11.6%) | 250 (13.8%) | 340 (15.0%) | 477 (14.0%) | 2052 (14.0%) | 361 (17.6%) |
| 35–39 | 372 (13.5%) | 336 (16.0%) | 335 (14.8%) | 264 (14.6%) | 327 (14.4%) | 486 (14.2%) | 2120 (14.5%) | 374 (17.6%) |
| 40–44 | 248 (9.0%) | 266 (12.7%) | 318 (14.1%) | 212 (11.7%) | 303 (13.4%) | 441 (12.9%) | 1788 (12.2%) | 268 (15.0%) |
| 45–49 | 198 (7.2%) | 174 (8.3%) | 236 (10.4%) | 151 (8.4%) | 198 (8.7%) | 318 (9.3%) | 1275 (8.7%) | 219 (17.2%) |
| 50–54 | 177 (6.4%) | 125 (6.0%) | 175 (7.7%) | 113 (6.3%) | 142 (6.3%) | 215 (6.3%) | 947 (6.5%) | 160 (16.9%) |
| 55+ | 363 (13.1%) | 153 (7.3%) | 201 (8.9%) | 250 (13.8%) | 95 (4.2%) | 514 (15.0%) | 1576 (10.8%) | 226 (14.3%) |
| **Gender** | | | | | | | | |
| Male | 1061 (38.4%) | 744 (35.5%) | 809 (35.7%) | 653 (36.1%) | 821 (36.2%) | 1250 (36.6%) | 5338 (36.5%) | 1,250 (23.4%) |
| Female | 1701 (61.6%) | 1354 (64.5%) | 1454 (64.3%) | 1154 (63.9%) | 1447 (63.8%) | 2168 (63.4%) | 9278 (63.5%) | 906 (9.8%) |
| **Marital status** | | | | | | | | |
| Single | 1203 (43.6%) | 355 (16.9%) | 1187 (52.5%) | 330 (18.3%) | 602 (26.5%) | 612 (17.9%) | 4289 (29.3%) | 611 (14.2%) |
| Married | 1167 (42.3%) | 1217 (58.0%) | 813 (35.9%) | 968 (53.6%) | 1173 (51.7%) | 1956 (57.2%) | 7294 (49.9%) | 1,081 (14.8%) |
| Widowed | 252 (9.1%) | 199 (9.5%) | 106 (4.7%) | 216 (12.0%) | 187 (8.2%) | 516 (15.1%) | 1476 (10.1%) | 169 (11.4%) |
| Divorced | 140 (5.1%) | 327 (15.6%) | 157 (6.9%) | 293 (16.2%) | 306 (13.5%) | 334 (9.8%) | 1557 (10.7%) | 295 (18.9%) |
| **HIV/ART status** | | | | | | | | |
| Negative | 837 (30.3%) | 503 (24.0%) | 741 (32.7%) | 611 (33.8%) | 696 (30.7%) | 936 (27.4%) | 4324 (29.6%) | 622 (14.4%) |
| HIV+ not on ART | 423 (15.3%) | 453 (21.6%) | 212 (9.4%) | 538 (29.8%) | 592 (26.1%) | 711 (20.8%) | 2929 (20.0%) | 587 (20%) |
| HIV+ On ART | 1502 (54.4%) | 1142 (54.4%) | 1310 (57.9%) | 658 (36.4%) | 980 (43.2%) | 1771 (51.8%) | 7363 (50.4%) | 947 (12.9%) |
| **Area of residence** | | | | | | | | |
| Rural | 2117 (76.6%) | 1100 (52.4%) | 1441 (63.7%) | 1084 (60.0%) | 973 (42.9%) | 2378 (69.6%) | 9093 (62.2%) | 1,199 (13.2%) |
| Urban | 645 (23.4%) | 998 (47.6%) | 822 (36.3%) | 723 (40.0%) | 1295 (57.1%) | 1040 (30.4%) | 5523 (37.8%) | 957 (17.3%) |
| **Highest Education level** | | | | | | | | |
| None | 211 (7.6%) | 217 (10.3%) | 207 (9.1%) | 370 (20.5%) | 92 (4.1%) | 154 (4.5%) | 1251 (8.6%) | 207 (16.5%) |
| Primary | 1033 (37.4%) | 1170 (55.8%) | 831 (36.7%) | 1175 (65.0%) | 889 (39.2%) | 1239 (36.2%) | 6337 (43.4%) | 885 (14.0%) |
| Secondary | 886 (32.1%) | 626 (29.8%) | 1175 (51.9%) | 23 (1.3%) | 1107 (48.8%) | 1882 (55.1%) | 5699 (39.0%) | 858 (15.1%) |
| Tertiary | 632 (22.9%) | 85 (4.1%) | 50 (2.2%) | 239 (13.2%) | 180 (7.9%) | 143 (4.2%) | 1329 (9.1%) | 206 (15.5%) |
| **Wealth index** | | | | | | | | |
| Poorest | 735 (26.6%) | 248 (11.8%) | 794 (35.1%) | 337 (18.6%) | 230 (10.1%) | 871 (25.5%) | 3215 (22.0%) | 492 (15.3%) |
| Poor | 625 (22.6%) | 244 (11.6%) | 587 (25.9%) | 357 (19.8%) | 282 (12.4%) | 703 (20.6%) | 2798 (19.1%) | 414 (14.8%) |
| Middle | 617 (22.3%) | 294 (14.0%) | 498 (22.0%) | 499 (27.6%) | 488 (21.5%) | 624 (18.3%) | 3020 (20.7%) | 413 (13.7%) |
| Rich | 462 (16.7%) | 397 (18.9%) | 293 (12.9%) | 369 (20.4%) | 636 (28.0%) | 632 (18.5%) | 2789 (19.1%) | 441 (15.8%) |
| Richest | 323 (11.7%) | 915 (43.6%) | 91 (4.0%) | 245 (13.6%) | 632 (27.9%) | 588 (17.2%) | 2794 (19.1%) | 396 (14.2%) |
| **Relationship to the household head** | | | | | | | | |
| Head | 1276 (46.2%) | 1118 (53.3%) | 1007 (44.5%) | 830 (45.9%) | 885 (39.0%) | 1753 (51.3%) | 6869 (47.0%) | 1,177 (17.1%) |
| Spouse | 410 (14.8%) | 511 (24.4%) | 298 (13.2%) | 451 (25.0%) | 615 (27.1%) | 664 (19.4%) | 2949 (20.2%) | 336 (11.4%) |
| Child | 537 (19.4%) | 271 (12.9%) | 391 (17.3%) | 262 (14.5%) | 410 (18.1%) | 439 (12.8%) | 2310 (15.8%) | 282 (12.2%) |
| Other | 539 (19.5%) | 198 (9.4%) | 567 (25.1%) | 264 (14.6%) | 358 (15.8%) | 562 (16.4%) | 2488 (17.0%) | 361 (14.5%) |

ART = Antiretroviral therapy, HIV = human immunodeficiency virus

[a]Overall is the denominator and specifically

[b]Numerator and the percentage is row percentage

and household heads than children (see Table 1). There was variation in harmful alcohol use by country ranging from 8.7% in Malawi to 26.1% in Namibia (see Fig 1).

**Factors associated with harmful alcohol use.** The factors associated with harmful alcohol use are shown in Table 2. Age, sex, residence, HIV/ART status, wealth index, and marital status were the key determinants of harmful alcohol use. The persons that were aged at least 20 years were more likely to use alcohol in a harmful way than those aged between 15 and 19 years. The females were less likely to use alcohol in a harmful way than the males (aOR: 0.32, 95%CI: 0.29–0.35, P<0.001). The urban residents were more likely to use alcohol in a harmful way than their rural counterparts (aOR: 1.74, 95%CI: 1.54–1.98, P<0.001). The individuals that were HIV positive and not yet on ART had similar high harmful alcohol use to those with a negative HIV test. However, the individuals that were HIV positive and on ART had a lower likelihood of harmful alcohol use (aOR: 0.65, 95%CI: 0.57–0.73, P<0.001). There was a declining trend in the likelihood of harmful alcohol use by wealth of individuals (see Table 2). Although those divorced had a similar high likelihood of harmful alcohol use to those single, those either married or widowed had lower likelihood of harmful alcohol use than those who were single (see Table 2).

## Predictive modelling of harmful alcohol use

Based on the predicted regression model, the key determinants of the harmful alcohol use were: gender, area of residence, HIV/ART status, marital status, and country of residence. After training the data, we used Super Learner, Decision Tree, Random Forest (RF), Lasso Regression, Sample mean and Gradient boosting on the test data and the model performance results are shown in Fig 2. Based on these models, the best performing models were Lasso or Super Learner or Random Forest were the best performing models while gradient boosting models or sample mean did not perform well. We settled for the Lasso model since it had the least value similar to the Super Learner.

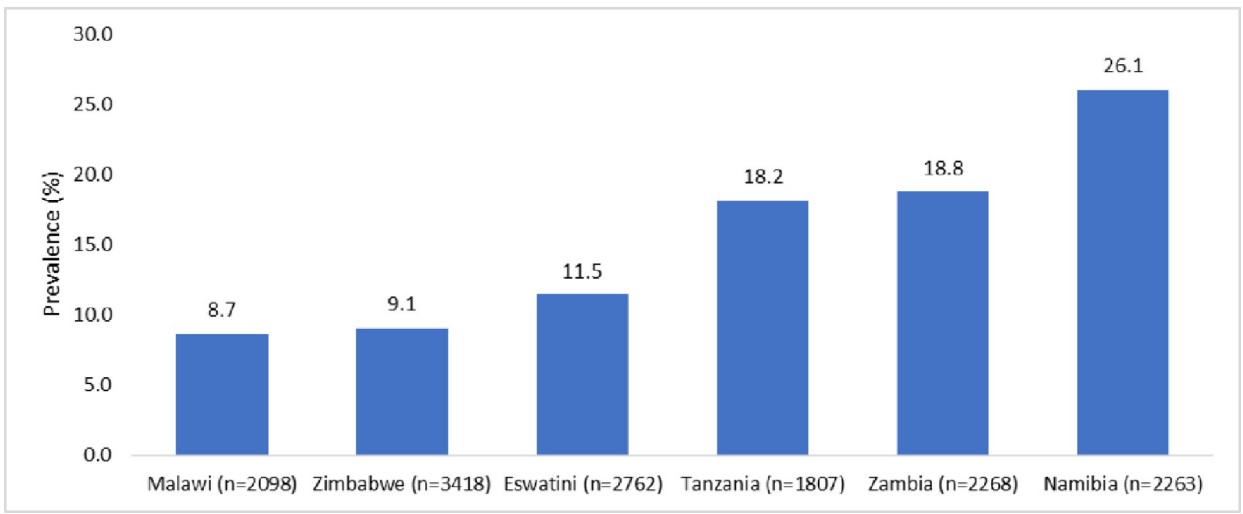

**Fig 1. Prevalence of harmful alcohol use in Eswatini, Malawi, Namibia, Tanzania, Zambia and Zimbabwe between 2015 and 2017.**

**Table 2. Logistic regression analysis of harmful alcohol use in in Eswatini, Malawi, Namibia, Tanzania, Zambia and Zimbabwe between 2015 and 2019.**

| Characteristics | Unadjusted | | | Adjusted | | |
|---|---|---|---|---|---|---|
| | Odds Ratio | 95%CI | P-value | Odds Ratio | 95%CI | P-value |
| **Age groups** | | | | | | |
| 15–19 | Ref | | | Ref | | |
| 20–24 | 2.87 | 2.20, 3.78 | <0.001 | 3.97 | 3.01, 5.27 | <0.001 |
| 25–29 | 4.48 | 3.48, 5.82 | <0.001 | 8.33 | 6.33, 11.0 | <0.001 |
| 30–34 | 4.43 | 3.46, 5.72 | <0.001 | 9.00 | 6.83, 12.0 | <0.001 |
| 35–39 | 4.44 | 3.48, 5.73 | <0.001 | 9.07 | 6.87, 12.1 | <0.001 |
| 40–44 | 3.65 | 2.84, 4.76 | <0.001 | 7.37 | 5.53, 9.91 | <0.001 |
| 45–49 | 4.3 | 3.31, 5.64 | <0.001 | 8.59 | 6.38, 11.6 | <0.001 |
| 50–54 | 4.21 | 3.19, 5.60 | <0.001 | 8.45 | 6.19, 11.6 | <0.001 |
| 55+ | 3.47 | 2.68, 4.54 | <0.001 | 6.77 | 5.06, 9.14 | <0.001 |
| **Gender** | | | | | | |
| Male | Ref | | | Ref | | |
| Female | 0.35 | 0.32, 0.39 | <0.001 | 0.32 | 0.29, 0.35 | <0.001 |
| **Area of residence** | | | | | | |
| Rural | Ref | | | Ref | | |
| Urban | 1.38 | 1.26, 1.51 | <0.001 | 1.74 | 1.54, 1.98 | <0.001 |
| **HIV/ART status** | | | | | | |
| Negative | Ref | | | Ref | | |
| HIV+ not on ART | 1.49 | 1.32, 1.69 | <0.001 | 1.06 | 0.92, 1.22 | 0.4 |
| HIV+ On ART | 0.88 | 0.79, 0.98 | 0.02 | 0.65 | 0.57, 0.73 | <0.001 |
| **Wealth index** | | | | | | |
| Poorest | Ref | | | Ref | | |
| Poor | 0.96 | 0.83, 1.11 | 0.6 | 0.87 | 0.75, 1.01 | 0.061 |
| Middle | 0.88 | 0.76, 1.01 | 0.07 | 0.72 | 0.62, 0.83 | <0.001 |
| Rich | 1.04 | 0.90, 1.20 | 0.6 | 0.71 | 0.60, 0.84 | <0.001 |
| Richest | 0.91 | 0.79, 1.05 | 0.2 | 0.58 | 0.48, 0.69 | <0.001 |
| **Marital status** | | | | | | |
| Single | Ref | | | Ref | | |
| Married | 1.05 | 0.94, 1.17 | 0.4 | 0.6 | 0.52, 0.68 | <0.001 |
| Widowed | 0.78 | 0.65, 0.93 | 0.007 | 0.7 | 0.56, 0.86 | <0.001 |
| Divorced | 1.41 | 1.21, 1.64 | <0.001 | 0.96 | 0.81, 1.15 | 0.7 |
| **Relationship to the household head** | | | | | | |
| Head | Ref | | | | | |
| Spouse | 0.62 | 0.55, 0.71 | <0.001 | | | |
| Child | 0.67 | 0.58, 0.77 | <0.001 | | | |
| Other | 0.82 | 0.72, 0.93 | 0.002 | | | |
| **Highest Education level** | | | | | | |
| None | Ref | | | | | |
| Primary | 0.82 | 0.70, 0.97 | 0.018 | | | |
| Secondary | 0.89 | 0.76, 1.06 | 0.2 | | | |
| Tertiary | 0.93 | 0.75, 1.14 | 0.5 | | | |

95%CI = 95% Confidence Interval, ART = Antiretroviral therapy, HIV = human immunodeficiency virus

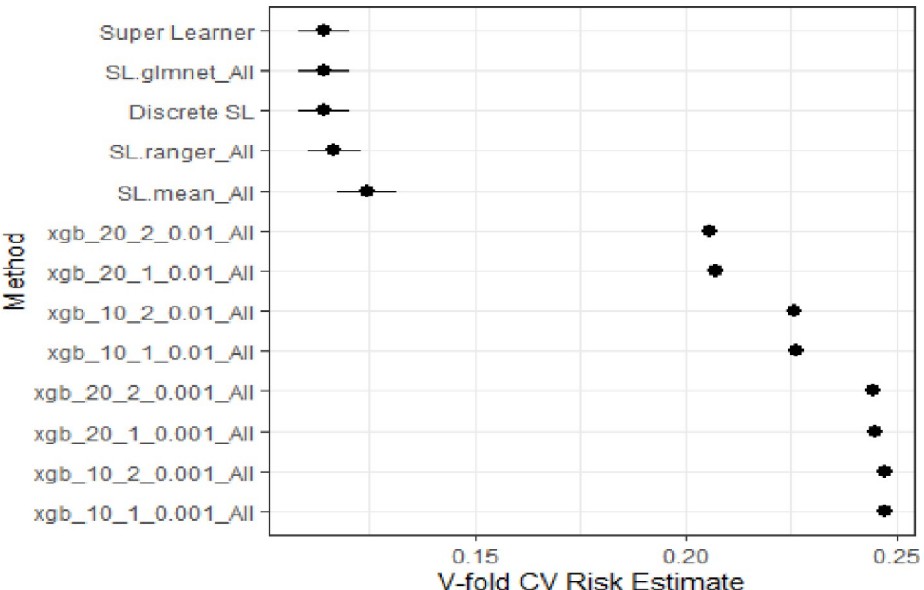

**Fig 2. Plot the performance for the different supervised machine learning models fitted.** Super Learner = Weighted average of the learners, SL.ranger_All = Random Forest, SL.glmnet_All = Lasso regression, SL.mean_All = Sample mean, Xgb_* = XGBoost algorithms, CV = Cross validation of the models.

## Assessment of the lasso model

The metrics for assessing the lasso regression model are shown in Table 3, The model exhibited high accuracy in the test and training datasets was 0.85. The precision for the model was 52%. The F1 Score was 2%.

## Predicted multiple logistic regression parameters

Fig 3 shows the predictors of harmful alcohol use based on test dataset for the Lasso/Logistic Regression. The output was consistent with the conventional data analysis shown in Table 2. The odds were found to increase with age, with those between the ages of 30 and 34 showing the highest risk. The predictions for females' odds were substantially lower than those for males, suggesting a protective effect, and these differences were significant (P<0.0001). Living in an urban area was linked to an increased risk of harmful alcohol use, with those who live there expected to have a higher chance of consuming alcohol in a harmful way than individuals who live in rural areas. Those who were HIV-positive and not receiving ART were at a higher risk of engaging in harmful alcohol use. Additionally, divorced individuals had a higher of alcohol consuming alcohol compared to those in marriage and the single.

**Table 3. Performance metrics for predicting harmful alcohol use in Eswatini, Malawi, Namibia, Tanzania, Zambia and Zimbabwe between 2015 and 2017.**

| Parameter | Value |
| --- | --- |
| Precision | 0.52 |
| Accuracy | 0.85 |
| Recall | 0.01 |
| False Positive Rate | 0.01 |
| False Negative Rate | 0.95 |
| F1 Score | 0.02 |

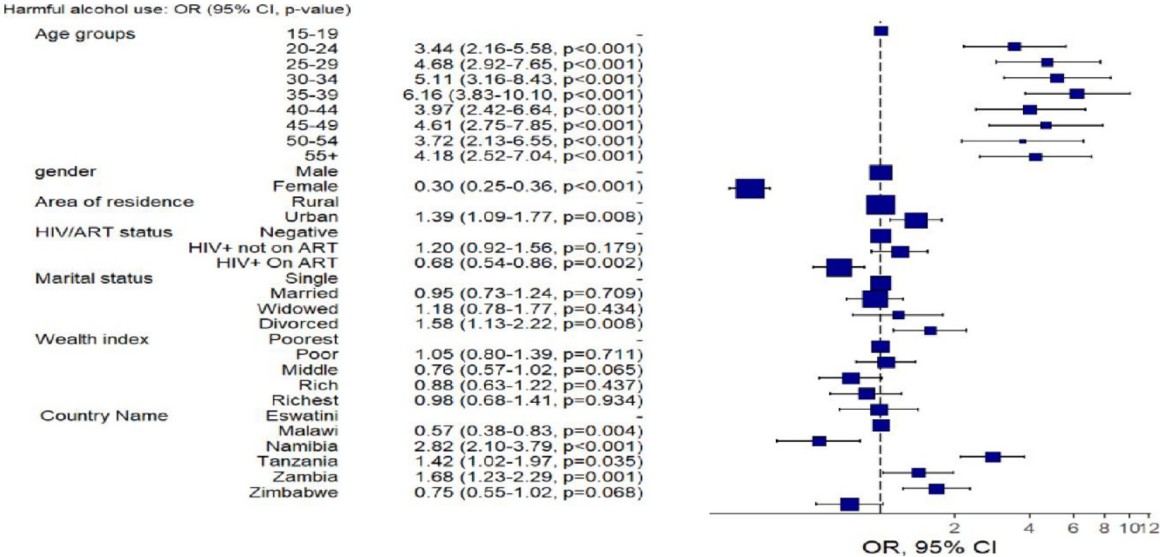

**Fig 3. Predictive model of the outcomes using logistic regression and the forest plot of odds ratio of harmful alcohol use in Eswatini, Malawi, Namibia, Tanzania, Zambia and Zimbabwe between 2015 and 2019.** OR = Odds ratio, 95%CI = 95% Confidence Interval, ART = Antiretroviral therapy, HIV = human immunodeficiency virus.

## Discussion

The study provided valuable insights into the prevalence of harmful alcohol consumption in six countries from 2015 to 2019. Namibia had the highest prevalence of harmful alcohol consumption at 18.5%, representing a significant public health problem. In contrast, Ethiopia had a lower prevalence of harmful alcohol consumption of 7.2%, suggesting possible cultural or contextual factors influencing alcohol consumption behavior. This finding resonates with existing literature highlighting elevated alcohol consumption rates in certain sub-Saharan African regions, emphasizing the need for targeted country interventions to address this pressing issue [17]. The interventions may target strengthening, reviewing and amendment of alcohol regulatory measures in these countries [18].

Consistent with earlier studies, the demographic analysis revealed specific patterns associated harmful alcohol consumption. Studies on alcohol consumption have consistently shown that age increases the odds of alcohol consumption. Younger age groups, particularly those between the ages of 20–24 showed lower odds, highlighting how vulnerable older age groups are. This is consistent with the trends identified in the literature, where risky drinking tends to peak in middle age [19, 20].

The study's findings regarding gender disparities validate the widespread belief that men are more likely than women to use harmful alcohol [6, 20]. Gender-sensitive interventions are crucial, as evidenced by the protective effect that females display, with their odds reduced by 12.3%. Contrary to literature highlighting a negative association between education and alcohol use, this study did not find significant associations. This deviation may be attributed to cultural differences and warrants further investigation.

The individuals on Antiretroviral Therapy (ART) displayed a decreased risk of harmful alcohol use, with odds reduced by 30.2%. This aligns with studies suggesting a complex interplay between HIV management and substance use [21]. The protective effect of ART emphasizes the potential dual benefits of HIV treatment programs in addressing both infectious and behavioral health challenges. Furthermore, this necessitates the need to routinely screen for

alcohol or other behavioral factors for the NCD in the routine HIV programmes to further propagate the benefits of the effect of HIV programme on NCD risk reduction.

The middle-income individuals exhibited lower odds of harmful alcohol use (14.8% lower odds), indicating potential disparities across socioeconomic status on harmful alcohol consumption. This aligns with research showing how socioeconomic status influences alcohol consumption patterns [22]. Variations by country, especially Namibia's higher odds, highlight how crucial it is to customize interventions to unique sociocultural contexts. However, the obtained harmful alcohol consumption is higher than what has been observed in other settings like Malawi [23]. The difference in the prevalence may be attributed to the study designs used. There was a strong correlation between the use of harmful alcohol and living in an urban area, with those who live in urban areas having higher odds. This inequality between urban and rural areas is consistent with the association between urbanization and alcohol consumption reported in several studies [7, 23]. The influence of marital status was also significant; married individuals demonstrated a protective effect, while divorced individuals showed increased odds (29.8% higher odds). These findings underscore the need for targeted interventions addressing the unique vulnerabilities of urban populations and divorced individuals.

Machine learning models provided valuable predictive insights into harmful alcohol use. The established logistic regression equation used in the model demonstrated high accuracy in identifying at-risk individuals. This analysis underscored the significance of age, gender, and HIV/ART status in predicting harmful alcohol use, aligning with the findings from traditional statistical analyses.

The study had a strong multi-country representation by using six different countries with varying sociocultural contexts. This inclusiveness improved the findings' generalizability to different social contexts. The study took a comprehensive approach by incorporating demographic variables with health-related factors like HIV/ART status. This enhanced understanding of the causes and effects of harmful alcohol consumption in the SSA context. This study was noteworthy for its innovative use of machine learning model. The study's conclusions are useful outside of the academic setting because they provide policymakers and public health professionals with doable suggestions. The study's reference period, covering data collected between 2015 and 2019, might not capture more recent developments in harmful alcohol consumption patterns.

In conclusion, this study used a multi-country approach to investigate harmful alcohol use in Sub-Saharan African countries using PHIA surveys conducted between 2015 and 2019. The research aimed to identify the factors associated with harmful alcohol use. With significant differences between the six countries under study, the prevalence findings highlighted the serious public health threat that harmful alcohol use poses. Particularly for Namibia, the prevalence rate of 18.5% highlights the need for focused interventions. Malawi, on the other hand, showed a prevalence rate that was lower at 8.7%, suggesting that cultural or contextual factors may have an impact on patterns of alcohol consumption. The complex relationships between age, gender, marital status, and education with harmful alcohol use were examined through demographic analyses. The need for age tailored interventions is reinforced by the age-specific analysis, where older age groups showed higher odds. The need for gender-sensitive approaches is highlighted by gender disparities, where men showed higher odds. Unexpectedly, the study did not find a substantial link between drinking alcohol negatively and education, indicating the need for more research. The analysis gained a new perspective with the inclusion of health-related variables, specifically HIV/ART status.

## Acknowledgments

The authors would like to thank the ICAP at Columbia University for allowing us to use the data.

## Author Contributions

**Conceptualization:** Mtumbi Goma, Wingston Felix Ng'ambi.

**Data curation:** Wingston Felix Ng'ambi.

**Formal analysis:** Wingston Felix Ng'ambi.

**Methodology:** Mtumbi Goma, Wingston Felix Ng'ambi.

**Supervision:** Cosmas Zyambo.

**Visualization:** Mtumbi Goma.

**Writing – original draft:** Mtumbi Goma.

**Writing – review & editing:** Cosmas Zyambo.

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
