## [Decision Letter · Decision Letter 0]

6 May 2024

PONE-D-24-10318Predicting Harmful Alcohol Use Prevalence in Sub-Saharan Africa between 2015 and 2019: Evidence from Population-based HIV Impact AssessmentPLOS ONE

Dear Dr. Goma,

Thank you for submitting your manuscript to PLOS ONE. After careful consideration, we feel that it has merit but does not fully meet PLOS ONE’s publication criteria as it currently stands. Therefore, we invite you to submit a revised version of the manuscript that addresses the points raised during the review process.

**ACADEMIC EDITOR:**

I would like to commend you on the thoroughness and rigor of your study. Your investigation into the prevalence of harmful alcohol use in Sub-Saharan Africa provides valuable insights into a critical public health issue.

However, there are minor revisions that need to be addressed:

**Background and Method Sections:** Clearly define harmful alcohol use in both the background and method sections to ensure clarity for readers.

**Figure and Table Titles:** Ensure that figure titles are placed below the figures, while table titles should be positioned above the tables for consistency and readability.

**Machine Learning Model Validation:** Describe whether the machine learning model used in your analysis was validated. Additionally, indicate any limitations associated with the model used for this analysis.

**Terminology:** Ensure consistency in terminology usage. For instance, use "HIV" or “HIV/AIDS” instead of "HIV/ART".

**Definition of Excessive Alcohol Use:** Clearly define what constitutes excessive alcohol use to provide readers with a clear understanding of this important terminology in your study.

**Inclusion of Ethiopia in the Study:** Acknowledge the absence of Ethiopia in the study population and discuss the implications of this omission. Additionally, if your study included findings on harmful alcohol use in Ethiopia, ensure that this is accurately reflected in the manuscript.

Please submit your revised manuscript by Jun 20 2024 11:59PM. If you will need more time than this to complete your revisions, please reply to this message or contact the journal office at plosone@plos.org. Please include the following items when submitting your revised manuscript:A rebuttal letter that responds to each point raised by the academic editor and reviewer(s). You should upload this letter as a separate file labeled 'Response to Reviewers'.A marked-up copy of your manuscript that highlights changes made to the original version. You should upload this as a separate file labeled 'Revised Manuscript with Track Changes'.An unmarked version of your revised paper without tracked changes. You should upload this as a separate file labeled 'Manuscript'.If applicable, we recommend that you deposit your laboratory protocols in protocols.io to enhance the reproducibility of your results. Protocols.io assigns your protocol its own identifier (DOI) so that it can be cited independently in the future. For instructions see: https://journals.plos.org/plosone/s/submission-guidelines#loc-laboratory-protocols. Additionally, PLOS ONE offers an option for publishing peer-reviewed Lab Protocol articles, which describe protocols hosted on protocols.io. Read more information on sharing protocols at https://plos.org/protocols?utm_medium=editorial-email&utm_source=authorletters&utm_campaign=protocols.

We look forward to receiving your revised manuscript.

Kind regards,

Yimam Getaneh Misganie, PhD

Academic Editor

PLOS ONE

Journal Requirements:

Reviewers' comments:

Reviewer's Responses to Questions

**Comments to the Author**

1. Is the manuscript technically sound, and do the data support the conclusions?

Reviewer #1: Yes

2. Has the statistical analysis been performed appropriately and rigorously? 

Reviewer #1: Yes

3. Have the authors made all data underlying the findings in their manuscript fully available?

Reviewer #1: Yes

4. Is the manuscript presented in an intelligible fashion and written in standard English?

Reviewer #1: Yes

5. Review Comments to the Author

Reviewer #1: General Comments

1-The article would benefit from proofreading for grammatical errors, repetition, and clarity.

2-Consideration should be given to the structure and organization of the article to improve flow and coherence.

3- Citations should follow a consistent format throughout the article.

Major Comments

1-The abstract is repetitive, appearing three times with the same content. This should be corrected to provide a concise summary of the study objectives, methods, key findings, and implications.

2-The introduction provides a good overview of the significance of harmful alcohol use and its association with HIV/AIDS globally and in SSA. However, it could be more concise and focused.

3-It would be helpful to clearly state the specific objectives of the study at the end of the introduction.

4- - The methods section lacks clarity in terms of the exact methodology employed for data analysis. It mentions the use of chi-square tests, logistic regression, and machine learning methods, but it doesn't provide details on how these were applied.

5-Providing more information on the variables included in the analysis and justification for their selection would enhance the methodological clarity.

6-The data management process is described briefly; providing more detail on how missing data were handled and how variables were recoded would improve transparency.

7-It's important to specify the criteria used for harmful alcohol use classification based on AUDIT-C scores.

8-The section on ethical considerations should be moved to the beginning of the methods section.

9-The results section is detailed and provides a comprehensive overview of participant characteristics and prevalence rates. However, it's quite lengthy and could be condensed

10-The presentation of results would benefit from clear subheadings and tables to improve readability.

11-The discussion of factors associated with harmful alcohol use could be more structured, perhaps separated into subsections based on demographic and health-related variables

12-The discussion should focus on interpreting the findings in relation to existing literature, explaining the implications of the results, and discussing limitations and future research directions.

13-There's an opportunity to explore potential mechanisms underlying the observed associations and to discuss the implications for public health interventions-It would be helpful to compare the findings of this study with previous research on harmful alcohol use and HIV/ART status in SSA.

14- The Authors should summarize the key findings of the study and their implications for public health practice and policy and Suggestions for future research based on the study's limitations and unanswered questions should be provided.

6. PLOS authors have the option to publish the peer review history of their article (what does this mean?). If published, this will include your full peer review and any attached files.

Reviewer #1: **Yes: **Sakarie Mustafe Hidig MD

---

## [Editor Report · Decision Letter 1]

30 Jul 2024

Predicting Harmful Alcohol Use Prevalence in Sub-Saharan Africa between 2015 and 2019: Evidence from Population-based HIV Impact Assessment

PONE-D-24-10318R1

Dear Dr. Mtumbi Goma, 

We’re pleased to inform you that your manuscript has been judged scientifically suitable for publication and will be formally accepted for publication once it meets all outstanding technical requirements.

Kind regards,

Yimam Getaneh Misganie (PhD, PhD)

Academic Editor

PLOS ONE

---

## [Editor Report · Acceptance letter]

5 Aug 2024

PONE-D-24-10318R1 

PLOS ONE

Dear Dr. Goma, 

I'm pleased to inform you that your manuscript has been deemed suitable for publication in PLOS ONE. Congratulations! Your manuscript is now being handed over to our production team.

Kind regards, 

on behalf of

Dr. Yimam Getaneh Misganie 

Academic Editor

PLOS ONE